# BIAS MITIGATION IN GRAPH DIFFUSION MODELS

**Meng Yu, Kun Zhan**[*]
School of Information Science & Engineering, Lanzhou University

## ABSTRACT

Most existing graph diffusion models have significant bias problems. We observe that the forward diffusion's maximum perturbation distribution in most models deviates from the standard Gaussian distribution, while reverse sampling consistently starts from a standard Gaussian distribution, which results in a reverse-starting bias. Together with the inherent exposure bias of diffusion models, this results in degraded generation quality. This paper proposes a comprehensive approach to mitigate both biases. To mitigate reverse-starting bias, we employ a newly designed Langevin sampling algorithm to align with the forward maximum perturbation distribution, establishing a new reverse-starting point. To address the exposure bias, we introduce a score correction mechanism based on a newly defined score difference. Our approach, which requires no network modifications, is validated across multiple models, datasets, and tasks, achieving state-of-the-art results.

## 1 INTRODUCTION

In recent years, graph diffusion models have made significant progress. GDSS (Jo et al., 2022) introduced the score-based diffusion model to the graph generative task, demonstrating remarkable results and outperforming existing baselines. This was followed by the development of more advanced graph diffusion models such as MOOD (Lee et al., 2023), GSDM (Luo et al., 2024), and HGDM (Wen et al., 2024). Due to the constraints imposed by the scale of graph data and the learning capacity of the networks, these models truncate the forward diffusion process to improve performance, preventing it from fully reaching the standard Gaussian distribution. However, in reverse sampling, they typically start from a standard Gaussian distribution without employing any specific strategy. We identify this mismatch as a critical bias issue. In addition, graph diffusion models also suffer from exposure bias, and we are working on addressing both of these challenges.

Diffusion models (Ho et al., 2020; Song et al., 2021) consist of a forward noising and a reverse denoising process. In the forward process, the data is gradually corrupted by noise over multiple steps. This process can be divided into four stages with the reduction of the signal-to-noise ratio: (1) the data distribution, (2) the low-noise stage, (3) the high-noise stage, and (4) the standard Gaussian.

**Reverse-Starting Bias.** Ideally, the forward process gradually perturbs the data to the standard Gaussian, while the reverse process starts from the standard Gaussian and gradually recovers clean data. However, in graph learning, due to limitations in data scale and the network's learning ability, it is difficult to accurately predict scores from the high-noise state or the standard Gaussian. This constrains the forward perturbation to follow a conservative strategy, where the maximum perturbation distribution deviates significantly from the standard Gaussian (Jo et al., 2022; Luo et al., 2024; Wen et al., 2024). Yet, the reverse-starting point remains the standard Gaussian in sampling, resulting in a severe reverse-starting bias as shown in Fig. 1, which significantly affects the generation quality.

**Exposure Bias.** During the forward process, the model generates a noisy sample $x_t$ based on a clean sample by adding noise. During the reverse process, the model starts from the standard Gaussian and iteratively denoises to obtain the predicted sample $\hat{x}_t$ using the score network. Due to the prediction error of the score network, this leads to exposure bias: a mismatch between $x_t$ in the forward diffusing and $\hat{x}_t$ in sampling. This bias accumulates and propagates as sampling progresses, ultimately affecting the quality of the generated samples. Naturally, the most direct approach to address exposure bias is to reduce the prediction errors of the score network.

---

[*]Corresponding Author. Email: kzhan@lzu.edu.cn

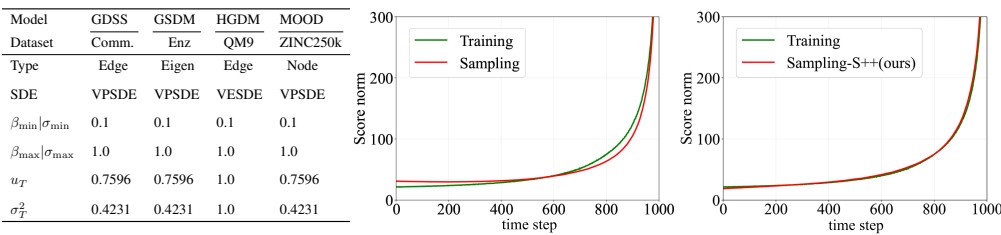

| Model | GDSS | GSDM | HGDM | MOOD |
|---|---|---|---|---|
| Dataset | Comm. | Enz | QM9 | ZINC250k |
| Type | Edge | Eigen | Edge | Node |
| SDE | VPSDE | VPSDE | VESDE | VPSDE |
| $\beta_{\min}\|\sigma_{\min}$ | 0.1 | 0.1 | 0.1 | 0.1 |
| $\beta_{\max}\|\sigma_{\max}$ | 1.0 | 1.0 | 1.0 | 1.0 |
| $u_T$ | 0.7596 | 0.7596 | 1.0 | 0.7596 |
| $\sigma_T^2$ | 0.4231 | 0.4231 | 1.0 | 0.4231 |

(a) Reverse-starting bias      (b) Baseline model-GDSS      (c) Our approach GDSS-S++

Figure 1: (a) According to the diffusion equation, we write the perturbation distribution as $q(\boldsymbol{x}_t|\boldsymbol{x}_0) = \mathcal{N}\left(\boldsymbol{x}_t|u_t\boldsymbol{x}_0, \sigma_t^2\mathbf{I}\right)$.The maximum perturbation distribution in the forward diffusing is $\mathcal{N}\left(\boldsymbol{x}_T|u_T\boldsymbol{x}_0, \sigma_T^2\mathbf{I}\right)$, but the reverse-starting point always follows $\mathcal{N}(\boldsymbol{x}_T|\boldsymbol{0}, \mathbf{I})$, leading to significant inconsistencies between forward diffusing and reverse sampling. In particular, we observe that $\boldsymbol{x}_T$ of different baselines are in the low-noise state since their signal-to-noise ratios are always greater than one. More details are provided in Appendix A. (b) and (c) Expectation of $\|\boldsymbol{s}_{\boldsymbol{\theta},t}(\cdot)\|_2$ during sampling (without the corrector) and training on Community-small. Due to the reverse-starting bias, there is a significant difference between forward diffusing and reverse sampling in the early stages of sampling in (b). However, after applying our approach, not only is the reverse-starting bias mitigated, but the exposure bias during the sampling process is also significantly reduced.

Rather than exploring the two biases independently, this paper aims to analyze and mitigate these two biases in graph diffusion models from a comprehensive perspective:

**Q1: Is it possible to mitigate exposure bias while addressing reverse-starting bias?** It originates from a key finding: when $\boldsymbol{x}_t$ is in the high-noise stage, the model is highly sensitive to the prediction error of the score network, which means the prediction error at this stage can significantly affect the generative quality. Conversely, when $\boldsymbol{x}_t$ is in the low-noise state, the model is quite resistant to the prediction error. Coincidentally, the forward maximum perturbation distribution of many models are in the low-noise state as shown in Fig. 1(a). Thus, for a given score network $s_{\boldsymbol{\theta},t}(\cdot)$, we use Langevin sampling (Song & Ermon, 2019) with $s_{\boldsymbol{\theta},T}(\cdot)$ to estimate the forward maximum perturbation distribution $q(\boldsymbol{x}_T|\boldsymbol{x}_0)$. It mitigates the reverse-starting bias, meanwhile, it pushes the reverse-starting point towards the low-noise state, utilizing the model's resistance to prediction error to avoid exposure bias.

However, the prediction error of the score network severely affects the stable distribution of Langevin sampling, forcing us to improve the prediction accuracy of the network, which is also beneficial for mitigating exposure bias in the sampling process. In particular, we also focus on the cost of achieving:

**Q2: How to correct scores without modifying the network or introducing other components?** The issue arises from a key situation: current graph diffusion models design different networks based on various domains, such as spatial, spectral, and hyperbolic domains. Our goal is to seamlessly integrate our correction method into these models without modifying the network architecture or model parameters. Additionally, we do not introduce any extra learner, such as a generator (Zheng et al., 2023) or a discriminator (Kim et al., 2023). Instead, we aim to fully leverage the existing diffusion model to address its inherent bias issues. First, we use the pretrained score network to generate a batch of samples. Second, we train a pseudo score network using these generated samples. Third, we use the score difference between the two networks to correct the score.

In summary, the main contributions of this paper are as follows: (1) To the best of our knowledge, we are the first to systematically address bias issues in graph diffusion models, effectively employing Langevin sampling to mitigate reverse-starting bias while significantly mitigating exposure bias in graph sampling. (2) We propose a score correction mechanism based on the score difference, and provide a logical analysis that the corrected scores are closer to the true scores, further mitigating reverse-starting bias and exposure bias. (3) Our approach does not require modifying the network or introducing new components. It has been validated on different graph diffusion models, different datasets, and different tasks, achieving state-of-the-art metrics.

## 2    RELATED WORK

Diffusion models were first introduced by Sohl-Dickstein et al. (2015) and later improved by Ho et al. (2020). Notably, Song et al. (2021) proposed a unified framework for diffusion models based on stochastic differential equations, greatly advancing their development. GDSS (Jo et al., 2022) was the first to introduce diffusion models to both nodes and edges of graphs. GSDM (Luo et al., 2024) extended GDSS by introducing the diffusion process of adjacency matrices into the spectral domain. HGDM (Wen et al., 2024) introduced node diffusion into hyperbolic space based on degree distribution characteristics. MOOD (Lee et al., 2023) propose the molecular out-Of-distribution diffusion to generate novel molecules with specific properties. Huang et al. (2023) proposed a conditional diffusion model based on discrete graph structures. Furthermore, Xu et al. (2022) proposed a graph diffusion model for predicting molecular conformations.

The signal leakage in image diffusion models is somewhat similar to the reverse-starting bias we found in graph diffusion models. It was first identified by Lin et al. (2024), who modified the diffusion noise schedule to ensure that the final time step in the forward process achieves a zero signal-to-noise Ratio. Then, Everaert et al. (2024) estimated the actual maximum perturbation distribution in the forward process as the reverse-starting point for sampling. The exposure bias of diffusion models was initially discovered in ADM-IP (Ning et al., 2023), which proposed re-perturbing the perturbation distribution to simulate exposure bias during sampling. EB-DDPM (Li & van der Schaar, 2023) estimated the upper bound of cumulative errors and incorporated it as a regularization term to retrain the model. MDSS (Ren et al., 2024) introduced a multi-step timed sampling strategy to mitigate exposure bias. Notably, ADM-IP, EB-DDPM, and MDSS all require retraining the model. In contrast, ADM-ES (Ning et al., 2024) proposed a noise scaling mechanism to mitigate exposure bias without retraining, while TS-DPM (Li et al., 2024) only requires finding the optimal time points during reverse sampling to match the forward process as closely as possible.

Our approach aims to effectively mitigate both reverse-starting bias and exposure bias without altering the network architecture or incorporating additional components.

## 3    MOTIVATION

### 3.1    GRAPH DIFFUSION MODELS

Firstly, we define a graph with $V$ nodes as $\boldsymbol{G} = (\boldsymbol{X}, \boldsymbol{A})$, where $\boldsymbol{X} \in \mathbb{R}^{V \times F}$ represents node features, with $F$ indicating that each node has $F$ features; $\boldsymbol{A} \in \mathbb{R}^{V \times V}$ represents the weighted adjacency matrix. Then, we formally represent the graph diffusion process as the trajectory of the random variable $\boldsymbol{G}$ over time $[0, T]$. The forward diffusion process is given by:

$$\mathrm{d}\boldsymbol{G}_t = \mathbf{f}_t(\boldsymbol{G}_t)\mathrm{d}t + g_t(\boldsymbol{G}_t)\mathrm{d}\boldsymbol{w}, \quad \boldsymbol{G}_0 \sim p_{\mathrm{data}}, \tag{1}$$

where $\mathbf{f}_t(\boldsymbol{G}_t)$ is the linear drift coefficient, $g_t(\boldsymbol{G}_t)$ is the diffusion coefficient, $\boldsymbol{w}$ is the standard Wiener process, and $\boldsymbol{G}_0$ is a graph from the true data distribution $p_{\mathrm{data}}$. The stochastic differential equation (SDE), Eq. (1), describes a forward diffusion process. Specifically, we replace $\boldsymbol{G}$ in Eq. (1) with node $\boldsymbol{X}$ or edge $\boldsymbol{A}$, representing the forward diffusion process of node $\boldsymbol{X}$ or edge $\boldsymbol{A}$, respectively.

Following GDSS (Jo et al., 2022), we separate $\boldsymbol{X}$ and $\boldsymbol{A}$ in the reverse diffusion:

$$\begin{aligned}
\mathrm{d}\boldsymbol{X}_t &= \left[\mathbf{f}_{1,t}(\boldsymbol{X}_t) - g_{1,t}^2 \nabla_{\boldsymbol{X}_t} \log p_t(\boldsymbol{X}_t, \boldsymbol{A}_t)\right] \mathrm{d}\bar{t} + g_{1,t}\mathrm{d}\bar{\boldsymbol{w}}_1, \\
\mathrm{d}\boldsymbol{A}_t &= \left[\mathbf{f}_{2,t}(\boldsymbol{A}_t) - g_{2,t}^2 \nabla_{\boldsymbol{A}_t} \log p_t(\boldsymbol{X}_t, \boldsymbol{A}_t)\right] \mathrm{d}\bar{t} + g_{2,t}\mathrm{d}\bar{\boldsymbol{w}}_2
\end{aligned} \tag{2}$$

where $\mathbf{f}_{1,t}$ and $\mathbf{f}_{2,t}$ satisfy $\mathbf{f}_t(\boldsymbol{X}, \boldsymbol{A}) = (\mathbf{f}_{1,t}(\boldsymbol{X}), \mathbf{f}_{2,t}(\boldsymbol{A}))$, representing the drift coefficients of the reverse process for $\boldsymbol{X}$ and $\boldsymbol{A}$ respectively. $g_{1,t}$ and $g_{2,t}$ are the corresponding scalar diffusion coefficients, $\bar{\boldsymbol{w}}_1$ and $\bar{\boldsymbol{w}}_2$ are reverse-time Wiener processes in reverse diffusion, and $\nabla_{\boldsymbol{X}_t} \log p(\boldsymbol{X}_t, \boldsymbol{A}_t)$ and $\nabla_{\boldsymbol{A}_t} \log p(\boldsymbol{X}_t, \boldsymbol{A}_t)$ represent the partial scores of the node and the edge, respectively. It's worth noting that two SDEs in Eq. (2) corresponds to the diffusion process of $\boldsymbol{X}$ and $\boldsymbol{A}$, respectively. We will choose different types of SDEs for $\boldsymbol{X}$ and $\boldsymbol{A}$ based on actual conditions. For example, for VPSDE (Variance-Preserving SDE) (Song et al., 2021), $\mathbf{f}_{1,t}(\boldsymbol{X}_t) = -\frac{1}{2}\beta_t \boldsymbol{X}_t$, $\mathbf{f}_{2,t}(\boldsymbol{A}_t) = -\frac{1}{2}\beta_t \boldsymbol{A}_t$, $g_{1,t} = g_{2,t} = \sqrt{\beta_t}$, $\beta_t = \beta_{\min} + t(\beta_{\max} - \beta_{\min})$, where $\beta_{\max}$ and $\beta_{\min}$ are hyperparameters.

Then, we use $\boldsymbol{s}_{\boldsymbol{\theta},t}(\boldsymbol{G}_t)$ and $\boldsymbol{s}_{\boldsymbol{\phi},t}(\boldsymbol{G}_t)$ to estimate the partial scores $\nabla_{\boldsymbol{X}_t} \log p(\boldsymbol{X}_t, \boldsymbol{A}_t)$ and $\nabla_{\boldsymbol{A}_t} \log p(\boldsymbol{X}_t, \boldsymbol{A}_t)$, respectively. Based on the idea of reverse denoising score matching, we derive

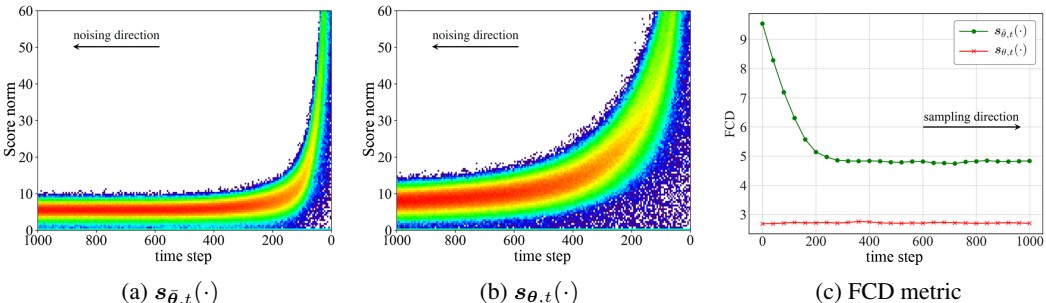

Figure 2: (a) and (b) The $\ell_2$ norm of the predictions from the two score networks at different time steps. (c) The response of the predictions of the two score networks to perturbations at different time steps in sampling.

$s_{\boldsymbol{\theta},t}(\boldsymbol{G}_t) \approx \nabla_{\boldsymbol{X}_t} \log p_{0t}(\boldsymbol{X}_t|\boldsymbol{X}_0)$ and $s_{\boldsymbol{\phi},t}(\boldsymbol{G}_t) \approx \nabla_{\boldsymbol{A}_t} \log p_{0t}(\boldsymbol{A}_t|\boldsymbol{A}_0)$, and the loss is:

$$
\min_{\boldsymbol{\theta}} \mathbb{E}_t \left\{ w_1(t) \mathbb{E}_{\boldsymbol{G}_0} \mathbb{E}_{\boldsymbol{G}_t|\boldsymbol{G}_0} \big\| s_{\boldsymbol{\theta},t}(\boldsymbol{G}_t) - \nabla_{\boldsymbol{X}_t} \log p_{0t}(\boldsymbol{X}_t|\boldsymbol{X}_0) \big\|_2^2 \right\},
$$
$$
\min_{\boldsymbol{\phi}} \mathbb{E}_t \left\{ w_2(t) \mathbb{E}_{\boldsymbol{G}_0} \mathbb{E}_{\boldsymbol{G}_t|\boldsymbol{G}_0} \big\| s_{\boldsymbol{\phi},t}(\boldsymbol{G}_t) - \nabla_{\boldsymbol{A}_t} \log p_{0t}(\boldsymbol{A}_t|\boldsymbol{A}_0) \big\|_2^2 \right\}
\tag{3}
$$

where $w_1(t)$ and $w_2(t)$ are positive weight functions, $t$ is uniformly sampled in the range of $[0, T]$. For nodes, we have $\boldsymbol{X}_0 \sim p_0(\boldsymbol{X})$, $\boldsymbol{X}_t \sim p_{0t}(\boldsymbol{X}_t|\boldsymbol{X}_0)$, and similarly for edges, we have $\boldsymbol{A}_0 \sim p_0(\boldsymbol{A})$, $\boldsymbol{A}_t \sim p_{0t}(\boldsymbol{A}_t|\boldsymbol{A}_0)$. Since $\mathbf{f}_{1,t}$ and $\mathbf{f}_{2,t}$ are affine, the transition kernels $p_{0t}(\boldsymbol{X}_t|\boldsymbol{X}_0)$ and $p_{0t}(\boldsymbol{A}_t|\boldsymbol{A}_0)$ are always Gaussian distributions, and closed-form means and variances are obtained based on standard techniques. For example, the node transition kernel in VPSDE (Song et al., 2021) is:

$$
p_{0t}(\boldsymbol{X}_t|\boldsymbol{X}_0) = \mathcal{N}\left( \boldsymbol{X}_t | e^{-\frac{1}{4}t^2(\beta_{\max}-\beta_{\min})-\frac{1}{2}t\beta_{\min}} \boldsymbol{X}_0, \mathbf{I} - \mathbf{I}e^{-\frac{1}{2}t^2(\beta_{\max}-\beta_{\min})-t\beta_{\min}} \right).
\tag{4}
$$

For simplicity, the subsequent derivations focus on $\boldsymbol{X}$ in VPSDE, as the derivations for both $\boldsymbol{X}$ and $\boldsymbol{A}$ in VPSDE and VESDE (Variance Exploding SDE) are similar to those for $\boldsymbol{X}$ in VPSDE.

## 3.2 WHY BASELINE MODELS ARE TRUNCATED?

In this section, we use GDSS as the basic model and QM9 (Ramakrishnan et al., 2014) as the dataset to demonstrate the reverse-starting bias and effectively validate our motivations. Specifically, we have two score networks: the first is a pretrained network $s_{\boldsymbol{\theta},t}(\cdot)$ whose forward maximum perturbation is far from reaching the standard Gaussian; the second is a network $s_{\bar{\boldsymbol{\theta}},t}(\cdot)$ whose forward maximum perturbation distribution is constrained to follow the standard Gaussian.

Figs. 2(a) and 2(b) show the $\ell_2$-norm distribution of the predictions of two score networks at different timesteps. Taking Fig. 2(a) as an example, we obtain perturbed samples through forward noising at each step, then use $s_{\bar{\boldsymbol{\theta}},t}(\cdot)$ to predict the score and calculate the corresponding $\ell_2$-norm value. We present the details of Fig. 2 in Appendix B. At time 0, the score $\ell_2$ norm of the ground-truth $\boldsymbol{X}_0$ spans $(0, 2500)$ approximately, demonstrating the diversity of the true data and its scores. As the noise intensity increases, the range of the score $\ell_2$ norm narrows, eventually stabilizing within $(0, 10)$. The evolution of the score $\ell_2$ norm of perturbed samples at different time indicates that as the distribution approaches the standard Gaussian distribution, the model becomes highly sensitive to score changes. The tightened score $\ell_2$ norm implies that the slight perturbation in score during the early sampling stage can significantly affect the generation quality. For Fig. 2(b), the evolution pattern of $s_{\boldsymbol{\theta},t}(\cdot)$ is consistent with $s_{\bar{\boldsymbol{\theta}},t}(\cdot)$, but since the maximum perturbation distribution of $s_{\boldsymbol{\theta},t}(\cdot)$ is far from reaching the standard Gaussian, its score $\ell_2$ norm range is wider, indicating a higher tolerance for score deviations.

Fig. 2(c) illustrates the response of the predictions of the two score networks to perturbations at different time steps in sampling. Each point in Fig. 2(c) corresponds to a perturbation experiment. The $x$-axis represents the addition of a standard Gaussian noise perturbation to the predicted score at the current time, while the $y$-axis represents the final generation metric resulting from that perturbation experiment (details in Appendix B). For $s_{\bar{\boldsymbol{\theta}},t}(\cdot)$, at time 0, we start from standard

Gaussian and perturb the predicted score at the current step. No perturbation is applied during the subsequent sampling steps, leading to a poor generation metric. As the time step of the perturbation increases, the negative impact on generation quality diminishes and eventually stabilizes after 200 steps. This evolution highlights how diffusion models are highly sensitive to accurate scores during the early sampling stages, where deviations in the score can significantly degrade the generated quality. For $s_{\theta,t}(\cdot)$, instead of starting from a standard Gaussian, we use samples from the forward maximum perturbation distribution as the initial point to eliminate the reverse-starting bias.

Fig. 2(c) shows that diffusion models are highly sensitive to score deviations in high-noise states, while in low-noise states, their resistance to score deviations significantly increases. Notably, this provides us with two potential approaches for addressing the reverse-starting bias: $s_{\bar{\theta},t}(\cdot)$ suggests retraining the model and constraining the forward maximum perturbation distribution to follow a standard Gaussian, while $s_{\theta,t}(\cdot)$ suggests exploring a reverse-starting distribution aligned with the forward maximum perturbation distribution during sampling. We observe that the latter not only mitigates the reverse-starting bias but also offers greater tolerance to subsequent deviations.

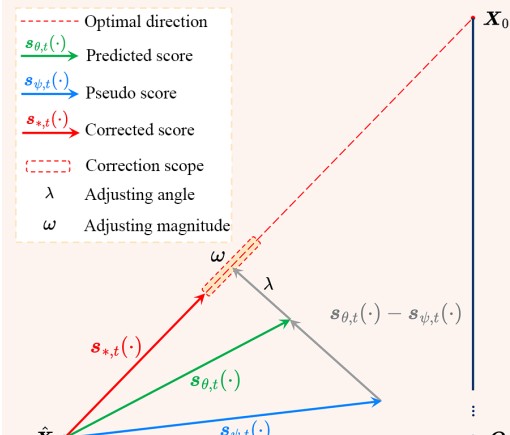

Figure 3: Score correction based on the score difference. At the reverse-sampling time $t$, the optimal score always points to $X_0$. $s_{\theta,t}(\cdot)$ points to $\gamma_t X_0$ with some deviation (partially containing $X_0$), while $s_{\psi,t}(\cdot)$ points to $\gamma'_t X_0$ with larger deviation (containing little $X_0$). The difference between predicted and pseudo scores guides the predicted score towards the optimal score. We use $\lambda$ to control the angle and $\omega$ to adjust the magnitude of the corrected score. The final corrected score flexibly approaches the optimal score within the dashed box.

## 4 METHODOLOGY

### 4.1 LANGEVIN SAMPLING

Langevin sampling is a key component of SDE-based diffusion models. Given sufficiently small step sizes and a large number of steps, Langevin sampling can utilize the score function to obtain samples from a probability distribution. Importantly, the prior distribution of Langevin sampling can be consistent with that of the diffusion model, typically a standard Gaussian distribution. Moreover, we already have a pretrained score network $s_{\theta,T}(\cdot) \approx \nabla_{X_T} \log q(X_T|X_0)$. This score guides Langevin sampling (Song & Ermon, 2019) to sample from the distribution $p(\hat{X}_T) \approx q(X_T|X_0)$:

$$\hat{X}_T \leftarrow \hat{X}_T + \frac{\eta_T}{2} s_{\theta}(\hat{X}_T, T) + \sqrt{\eta_T}\epsilon_T \tag{5}$$

where the subscript $T$ represents the reverse-starting time. In the reverse-starting alignment stage, we only use the score $s_{\theta,T}(\cdot)$ at time $T$. $\eta_T$ represents the step size at time $T$, and $\epsilon_T$ is standard Gaussian noise. After iterating Eq. (5) $M$ times, we assume the distribution $p(\hat{X}_T)$ has transitioned to a stable state. We refer to this stage as the reverse-starting alignment stage.

### 4.2 S++ BIAS CORRECTION METHOD

In theory, after Langevin sampling based on Eq. (5), $\hat{X}_T$ almost follows the distribution $q(X_T|X_0)$. However, the pretrained score $s_{\theta,T}(\cdot)$ struggles to accurately learn the true score $\nabla_{X_T} \log q(X_T|X_0) = \frac{X_T - \sqrt{\bar{\alpha}_T}X_0}{1-\bar{\alpha}_T}$ where $\bar{\alpha}_t = 1 - \beta_t, \forall t \in [0, T]$. We have to consider the exposure bias. Without loss of generality, we consider the predicted score at any time with

$$s_{\theta,t}(\hat{X}_t) = -\frac{\hat{X}_t - \sqrt{\bar{\alpha}_t}\hat{X}_0}{1-\bar{\alpha}_t} . \tag{6}$$

We can rewrite Eq. (6) as

$$\hat{X}_0 = \frac{1}{\sqrt{\bar{\alpha}_t}}\left(\hat{X}_t + (1-\bar{\alpha}_t)s_{\theta,t}(\hat{X}_t)\right) = \hat{X}_0(\hat{X}_t, \theta) . \tag{7}$$

Straightforwardly, it is challenging to predict $\hat{\boldsymbol{X}}_0$ from $\boldsymbol{X}_0$ analytically. Following Zhang et al. (2023), we model the estimate of $\hat{\boldsymbol{X}}_0$ by

$$\hat{\boldsymbol{X}}_0 = \gamma_t \boldsymbol{X}_0 + \delta_t \boldsymbol{\epsilon}_a \tag{8}$$

where $\gamma_t$ and $\delta_t$ are constants that reflect the similarity of $\hat{\boldsymbol{X}}_0$ to $\boldsymbol{X}_0$ and $\boldsymbol{\epsilon}_a$, respectively, with $\boldsymbol{\epsilon}_a \sim \mathcal{N}(\boldsymbol{0}, \boldsymbol{I})$. Consequently, for all $j, k$ such that $0 \leq j < k \leq N$, it holds that $1 > \gamma_j > \gamma_k \geq 0$ and $0 \leq \delta_j < \delta_k$. We then use the model $\boldsymbol{s}_{\boldsymbol{\theta},t}(\hat{\boldsymbol{X}}_t)$ to generate a dataset $\boldsymbol{X}_0'$ and train a pseudo score network $\boldsymbol{s}_{\boldsymbol{\psi},t}(\cdot)$ using $\boldsymbol{X}_0'$. Since $\hat{\boldsymbol{X}}_0$ always has a bias compared to $\boldsymbol{X}_0$, the pseudo score $\boldsymbol{s}_{\boldsymbol{\psi},t}(\cdot)$, trained on the pseudo data $\boldsymbol{X}_0'$, naturally learns the bias. Similarly, we estimate $\hat{\boldsymbol{X}}_0'(\hat{\boldsymbol{X}}_t, \boldsymbol{\psi})$ as $\hat{\boldsymbol{X}}_0' = \gamma_t' \boldsymbol{X}_0 + \delta_t' \boldsymbol{\epsilon}_b$, where it is straightforward that $\gamma_t' < \gamma_t$. Then, we define a score difference between the two scores at the same time point for the same sample:

$$\boldsymbol{s}_{\boldsymbol{\theta},t}(\hat{\boldsymbol{X}}_t) - \boldsymbol{s}_{\boldsymbol{\psi},t}(\hat{\boldsymbol{X}}_t) = \sqrt{\bar{\alpha}_t} \frac{(\gamma_t - \gamma_t')\boldsymbol{X}_0 + (\delta_t \boldsymbol{\epsilon}_a - \delta_t' \boldsymbol{\epsilon}_b)}{1 - \bar{\alpha}_t} . \tag{9}$$

We find that the score difference contains information about $\boldsymbol{X}_0$ as shown in Fig. 3. We aim to utilize this information. Inspired by classifier-free guidance (Ho & Salimans, 2021) and extrapolation operations (Zhang et al., 2023), we define a new score adjustment by

$$\begin{aligned} \boldsymbol{s}_{\boldsymbol{\theta},t}(\hat{\boldsymbol{X}}_t) &= \boldsymbol{s}_{\boldsymbol{\theta},t}(\hat{\boldsymbol{X}}_t) + \lambda\big(\boldsymbol{s}_{\boldsymbol{\theta},t}(\hat{\boldsymbol{X}}_t) - \boldsymbol{s}_{\boldsymbol{\psi},t}(\hat{\boldsymbol{X}}_t)\big) \\ &= -\frac{\hat{\boldsymbol{X}}_t - \sqrt{\bar{\alpha}_t}\Big(\big(\gamma_t + \lambda(\gamma_t - \gamma_t')\big)\boldsymbol{X}_0 + \delta_t \boldsymbol{\epsilon}_a + \lambda(\delta_t \boldsymbol{\epsilon}_a - \delta_t' \boldsymbol{\epsilon}_b)\Big)}{1 - \bar{\alpha}_t} \end{aligned} \tag{10}$$

where $\lambda \geq 0$ represents the step size for correcting the score using the score difference, Eq. (9). When $\lambda = 0$, no correction is applied. Conceptually, the correction operation pulls the biased direction towards the unbiased direction. Although there is some noise in this correction direction, choosing appropriate parameters $\lambda$ improves the score accuracy. Then, we divide the $\boldsymbol{s}_{\boldsymbol{\theta},t}(\hat{\boldsymbol{X}}_t)$ by a scalar to adjust the score magnitude, further driving $\boldsymbol{s}_{\boldsymbol{\theta},t}(\hat{\boldsymbol{X}}_t)$ closer to the true score:

$$\boldsymbol{s}_{\boldsymbol{\theta},t}(\hat{\boldsymbol{X}}_t) = \frac{\boldsymbol{s}_{\boldsymbol{\theta},t}(\hat{\boldsymbol{X}}_t)}{\omega} . \tag{11}$$

In particular, we emphasize that the S++ score correction at time $T$ is far more critical than at other times. In some cases, the correction at time $T$ is so effective that no further correction is needed in subsequent stages, especially in a truncated model. Therefore, we recommend decoupling the correction parameter at time $T$ from those at other times during the actual score correction process. Specifically, by using Langevin sampling (Song & Ermon, 2019) to align the distribution with the forward maximum perturbation distribution, which is in a low-noise state and retains some data information from $\boldsymbol{X}_0$, we can shorten the sampling chain and significantly reduce sampling time. Experimental validation is provided in §5.3.

The score correction method described here can also be applied in the reverse-starting alignment stage, as Langevin sampling also relies on scores. Once the reverse-starting bias is corrected, it naturally helps mitigate the exposure bias in subsequent sampling stages. Additionally, our score correction method is applicable wherever scores are utilized, ensuring its effectiveness across different stages of the reverse sampling.

We emphasize that utilizing Langevin sampling to obtain aligned samples and using the difference signal to correct scores are indispensable components for addressing the reverse-starting bias and the exposure bias. The effect is shown in Fig. 1(c). Additionally, we conduct extensive ablation experiments in §5.4 to demonstrate this point. We provide a detailed geometric illustration in Fig. 3 and provide detailed derivations and proofs of the equations from §4 in Appendix C.

## 5 EXPERIMENT

In this section, we select three generic graph datasets and two molecular datasets to evaluate the performance of our approach. In order to demonstrate the broad applicability of this method in addressing the reverse-starting bias and mitigating exposure bias, we tested it on a variety of

Table 1: Generation results on the generic graph datasets (Lower is better). The results of the Enzymes dataset of GDSS are reproduced by ourselves, the results of other baselines are all from published papers, and we give detailed settings and instructions in Appendix D.

| Dataset Info. | Community-small Synthetic, $12 \leq |V| \leq 20$ | | | | Enzymes Real, $10 \leq |V| \leq 125$ | | | | Grid Synthetic, $100 \leq |V| \leq 400$ | | | |
|---|---|---|---|---|---|---|---|---|---|---|---|---|
| Method | Deg.↓ | Clus.↓ | Orbit↓ | Avg.↓ | Deg.↓ | Clus.↓ | Orbit↓ | Avg.↓ | Deg.↓ | Clus.↓ | Orbit↓ | Avg.↓ |
| GDSS-OC | 0.050 | 0.132 | 0.011 | 0.064 | 0.052 | 0.627 | 0.249 | 0.309 | 0.270 | 0.009 | 0.034 | 0.070 |
| GDSS-OC-S++ | **0.021** | **0.061** | **0.005** | **0.029** | **0.067** | **0.099** | **0.007** | **0.058** | **0.105** | **0.004** | **0.061** | **0.066** |
| GDSS-WC | 0.045 | 0.088 | 0.007 | 0.045 | 0.044 | 0.069 | 0.002 | 0.038 | 0.111 | 0.005 | 0.070 | 0.070 |
| GDSS-WC-S++ | **0.019** | **0.062** | **0.004** | **0.028** | **0.031** | **0.050** | **0.003** | **0.028** | **0.105** | **0.004** | **0.061** | **0.057** |
| HGDM-OC | 0.065 | 0.119 | 0.024 | 0.069 | 0.125 | 0.625 | 0.371 | 0.374 | 0.181 | 0.019 | 0.112 | 0.104 |
| HGDM-OC-S++ | **0.021** | **0.034** | **0.005** | **0.020** | **0.080** | **0.500** | **0.225** | **0.268** | **0.023** | **0.034** | **0.004** | **0.020** |
| HGDM-WC | 0.017 | 0.050 | 0.005 | 0.024 | 0.045 | 0.049 | 0.003 | 0.035 | 0.137 | 0.004 | 0.048 | 0.069 |
| HGDM-WC-S++ | **0.021** | **0.024** | **0.004** | **0.016** | **0.040** | **0.041** | **0.005** | **0.029** | **0.123** | **0.003** | **0.047** | **0.058** |
| GSDM-OC | 0.142 | 0.230 | 0.043 | 0.138 | 0.930 | 0.867 | 0.168 | 0.655 | 1.996 | 0.0 | 1.013 | 1.003 |
| GSDM-OC-S++ | **0.011** | **0.016** | **0.001** | **0.009** | **0.012** | **0.087** | **0.011** | **0.037** | **1.2e-4** | **0.0** | **1.2e-4** | **0.066** |
| GSDM-WC | 0.011 | 0.016 | 0.001 | 0.009 | 0.013 | 0.088 | 0.013 | 0.038 | 0.002 | 0.0 | 0.0 | 7.2e-5 |
| GSDM-WC-S++ | **0.011** | **0.016** | **0.001** | **0.009** | **0.011** | **0.086** | **0.010** | **0.036** | **5.0e-5** | **0.0** | **1.1e-5** | **0.066** |

mainstream graph diffusion models, namely GDSS (Jo et al., 2022), GSDM (Luo et al., 2024), HGDM (Wen et al., 2024), and MOOD (Lee et al., 2023). Our improved model is prefixed with the basic diffusion model and denoted by S++ at its suffix. At the same time, we perform extensive downstream task testing and ablation study to further illustrate the effectiveness and necessity of S++.

## 5.1 GENERIC GRAPH GENERATION

**Experimental Setup** We select three generic graph datasets to test our approach: (1) Community-small: 100 artificially generated graphs with community structure; (2) Enzymes: 600 protein maps representing the enzyme structure in BRENDA (Schomburg et al., 2004); (3) Grid: 100 standard 2D grid diagrams. To evaluate the quality of the generated graphs, we follow GDSS (Jo et al., 2022) and we use Maximum Mean Difference (MMD) (Gretton et al., 2012; You et al., 2018) to compare the statistical distribution of the graphs between the same number of generated plots and the test plots, including the distribution of measured degrees, clustering coefficients, and the number of orbits of the 4-node track.

**Results** Table 1 shows that S++ outperforms all of baselines. For the uncorrected sampling method, the performance indicators of the baseline model are particularly poor due to the existence of the reverse-starting bias and score exposure bias, while S++ can significantly improve the performance of all baseline models and reach or even exceed the level of the baseline model with correctors. Because the method without correctors can significantly reduce computational complexity, we believe that S++ can really release the ability of the graph diffusion model, which is enlightening for large-scale datasets. For the sampling method with aligners, S++ is still significantly better than all baseline models. At the same time, we also give experimental comparisons of other advanced models in Appendix F, and results show that S++ achieves SOTA indicators of the corresponding tasks.

## 5.2 MOLECULAR GRAPH GENERATION

**Experimental Setup** We select two widely recognized molecular datasets to evaluate our approach: QM9 (Ramakrishnan et al., 2014) and ZINC250k (Irwin et al., 2012). We generate 10,000 molecules and select the following metrics: Frechet ChemNet Distance (FCD) (Preuer et al., 2018), Neighborhood subgraph pairwise distance kernel (NSPDK) MMD (Costa & Grave, 2010), validity w/o correction, and the generation time. (1) FCD uses the activation of the penultimate layer of ChemNet to calculate the distance between the benchmark molecular dataset and the generated dataset to characterize the similarity between them, and the lower the FCD, the higher the similarity between distributions. (2) NSPDK MMD considers the characteristics of nodes and edges at the same time, and calculates the MMD between the benchmark molecular dataset and the generated dataset; (3) Sampling time is used to evaluate the speed in generating large-scale molecular datasets, and we only count the time spent on sampling, regardless of the time spent on preprocessing and evaluation.

Table 2: Comparison of different methods on QM9 and ZINC250k datasets.

| Method | QM9 | | | ZINC250k | | |
|---|---|---|---|---|---|---|
| | Sampling time ↓ | NSPDK MMD ↓ | FCD ↓ | Sampling time↓ | NSPDK MMD ↓ | FCD ↓ |
| GDSS-OC | 0.73e2 | 0.016 | 4.584 | 0.73e3 | 0.047 | 20.53 |
| GDSS-OC-S++ | **5.10** | **0.001** | **1.661** | **0.70e3** | **0.050** | **16.79** |
| GDSS-WC | 1.61e2 | 0.004 | 2.550 | 1.41e3 | 0.019 | 14.66 |
| GDSS-WC-S++ | **9.25** | **0.001** | **1.661** | **0.98e3** | **0.012** | **12.70** |
| HGDM-OC | 0.62e2 | 0.005 | 3.164 | 0.76e3 | 0.033 | 21.38 |
| HGDM-OC-S++ | **0.62e2** | **0.003** | **2.512** | **0.77e3** | **0.034** | **20.79** |
| HGDM-WC | 1.16e2 | 0.002 | 2.147 | 1.52e3 | 0.016 | 17.69 |
| HGDM-WC-S++ | **0.98e2** | **0.001** | **2.001** | **1.17e3** | **0.016** | **16.24** |

**Results** Table 2 shows that both in terms of sampling time and generation quality, S++ is significantly better than the baseline models. For the sampling method without correctors, due to the existence of the reverse-starting bias and score exposure bias, the quality of generation from the baseline model is particularly poor, while S++ can significantly improve the performance of all baselines and approximate the sampling methods with correctors of the baseline model. For the sampling method with aligners, S++ is still significantly better than all baseline models and greatly reduces the sampling time. At the same time, we provide more comparative experimental results in Appendix F and provide parameter sensitivity experiments in Appendix G.

## 5.3 DIVERSITY GENERATION

**Characteristic molecule generation** To evaluate the performance of S++ in generating novel, drug-like, and synthesizable molecules, we follow (Lee et al., 2023) and assess S++ in the five docking score (DS) optimization tasks under the quantitative estimate of synthetic accessibility (SA), drug-likeness (QED) and novelty constraints. The property $Y$ is defined by

$$Y(\boldsymbol{G}) = \widehat{\text{DS}}(\boldsymbol{G}) \times \text{QED}(\boldsymbol{G}) \times \widehat{\text{SA}}(\boldsymbol{G}) \in [0, 1] \tag{12}$$

where $\widehat{\text{DS}}$ refers to the normalized docking score, $\widehat{\text{SA}}$ denotes the normalized synthetic accessibility, and QED represents drug-likeness. We use MOOD-S++ to generate 3000 molecules and evaluate performance using the following metrics. **Novel hit ratio (%)** is the fraction of unique hit molecules whose maximum Tanimoto similarity with the training molecules is less than 0.4. In particular, hit molecules are defined as the molecules that satisfy the following conditions: DS < (the median DS of the known active molecules), QED > 0.5, and SA < 5. **Novel top 5% docking score** refers to the average DS of the top 5% unique molecules that satisfy the constraints QED > 0.5 and SA < 5 and their maximum similarity with the training molecules is below 0.4. To avoid bias in target selection, we utilize five protein targets: parp1, fa7, 5ht1b, braf, and jak2.

**Results** Tables 3 and 4 show that MOOD-S++ is significantly better than baseline in all target proteins. This indicates that S++ still has advantages in the discovery of drug-like, synthesizable, and novel molecular tasks with high binding affinity, and it can be seen that the reverse-starting bias and exposure bias pose a significant threat to various generation tasks.

**Accelerate generation** To demonstrate that S++ can generate good samples faster by using fewer steps of reverse diffusion, we choose GDSS-OC as the benchmark, and QM9 and Comm are selected to test the performance of our approach and benchmark models at different sampling total time steps.

**Results** Table 5 shows that S++ is significantly better than the baseline model at different sampling total time steps $N$. S++ is not only able to generate samples with fewer reverse-diffusion steps but also achieves consistent improvements across generation metrics, especially on the QM9 dataset, where S++ remained close to optimal performance even with a significant reduction in the sampling time step ($N = 100$), while the performance of the benchmark model decreased significantly.

In conclusion, S++ shows higher efficiency, better quality, and stronger robustness in graph generative tasks, which provides a powerful improvement scheme for the application of the diffusion model.

Table 3: Novel hit ratio (%) results (↑).

| Method | Target protein | | | | |
|--------|-------|-----|------|------|------|
| | parp1 | fa7 | 5ht1b | braf | jak2 |
| MOOD | 7.017 (± 0.428) | 0.733 (± 0.141) | 18.673 (± 0.423) | 5.240 (± 0.285) | 9.200 (± 0.524) |
| MOOD-S++ | **8.286 (± 0.214)** | **0.900 (± 0.068)** | **20.354 (± 0.672)** | **5.653 (± 0.073)** | **9.167 (± 0.067)** |

Table 4: Novel top 5% docking score (kcal/mol) results (↓).

| Method | Target protein | | | | |
|--------|-------|-----|------|------|------|
| | parp1 | fa7 | 5ht1b | braf | jak2 |
| MOOD | -10.865 (± 0.113) | -8.160 (± 0.071) | -11.145 (± 0.042) | -11.063 (± 0.034) | -10.147 (± 0.060) |
| MOOD-S++ | **-10.961 (± 0.027)** | **-8.182 (± 0.028)** | **-11.231 (± 0.036)** | **-11.143 (± 0.025)** | **-10.163 (± 0.015)** |

Table 5: Comparison of different methods under different total sampling time steps.

| $N$ | Method | QM9 | | | Community-small | | | |
|------|--------|-----------------|-------------|--------|-------|---------|---------|--------|
| | | Val. w/o corr. ↑ | NSPDK MMD ↓ | FCD ↓ | Deg.↓ | Clus. ↓ | Orbit ↓ | Avg. ↓ |
| 1000 | GDSS-OC | 73.5 | 0.015 | 4.584 | 0.050 | 0.132 | 0.011 | 0.064 |
| | GDSS-OCS++ | **94.0** | **0.001** | **1.671** | **0.021** | **0.061** | **0.005** | **0.029** |
| 500 | GDSS-OC | 46.2 | 0.045 | 7.960 | 0.136 | 0.456 | 0.151 | 0.248 |
| | GDSS-OC-S++ | **93.9** | **0.001** | **1.665** | **0.029** | **0.142** | **0.008** | **0.060** |
| 100 | GDSS-OC | 37.8 | 0.069 | 9.951 | 0.092 | 0.666 | 0.394 | 0.384 |
| | GDSS-OC-S++ | **93.9** | **0.001** | **1.663** | **0.061** | **0.414** | **0.140** | **0.205** |

Table 6: Ablation experiments on the QM9 dataset.

| Method | QM9 | | |
|--------|-----------------|-------------|-------|
| | Val. w/o corr. ↑ | NSPDK MMD ↓ | FCD ↓ |
| GDSS-OC | 73.5 | 0.0157 | 4.58 |
| GDSS-w/o correction in sampling | 94.8 | 0.0037 | 2.65 |
| GDSS-w/o reverse-starting alignment | 89.8 | 0.0031 | 2.01 |
| GDSS-OC-S++ | **94.0** | **0.0014** | **1.67** |

## 5.4 ABLATION STUDY

Table 6 demonstrates that GDSS without reverse-starting alignment and score correction independently improves performance to varying degrees. Among these, the case without reverse-starting alignment is more general, as it does not require $\eta_T$ and $T$. Even in their absence, our approach still achieves impressive results, highlighting its robustness and flexibility. Additionally, we provide a comparative analysis of the two biases in the image and graph domains in Appendix H and present comparative experiments of S++ with existing methods on images in Appendix I.

## 6 CONCLUSION

In this paper, we use Langevin sampling to obtain samples aligned with the forward maximum perturbation distribution, which mitigates the reverse-starting bias and greatly mitigates the exposure bias of the score network, and we propose a score correction mechanism based on score difference to further promote the stable-state distribution of Langevin sampling to the true forward maximum perturbation distribution, and further mitigate the exposure bias in sampling. Since the score is also used in Langevin sampling, the proposed score correction is also applied to Langevin sampling as well. Our approach does not require network modifications or the introduction of new learners and can be naturally integrated into existing graph diffusion models to achieve SOTA metrics on multiple datasets and multiple tasks.

## ACKNOWLEDGMENTS

This work was supported by the National Natural Science Foundation of China under Grant No. 62176108.

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

## A    REVERSE-STARTING BIAS

In this section, we provide a detailed discussion of the reverse-starting bias. It is worth noting that these models are based on SDE (Song et al., 2021). For VPSDE, it obtains perturbed samples through Eq. (4), which corresponds to Eq. (33) in (Song et al., 2021). At $t = T$, Eq. (4) gives the maximum perturbation distribution, which is $\mathcal{N}(\boldsymbol{X}_T|\mathbf{0}, \mathbf{I})$. Similarly, for VESDE, the diffusion model obtains perturbed samples through $p_{0t}(\boldsymbol{X}_t|\boldsymbol{X}_0) = \mathcal{N}(\boldsymbol{X}_t|\boldsymbol{X}_0, \sigma_t^2\mathbf{I})$, where $\sigma_{\min} = \sigma_1 \leq \sigma_t \leq \sigma_T = \sigma_{\max}$. When this expression is extended continuously, we have

$$p_{0t}(\boldsymbol{X}_t|\boldsymbol{X}_0) = \mathcal{N}\Big(\boldsymbol{X}_t|\boldsymbol{X}_0, \sigma_{\min}^2\Big(\frac{\sigma_{\max}}{\sigma_{\min}}\Big)^{2t}\mathbf{I}\Big), t \in [0, 1]. \tag{13}$$

Eq. (13) corresponds to Eq. (31) of (Song et al., 2021). At $t = T$ Eq. (13) achieves the maximum perturbation distribution, which is $\mathcal{N}(\boldsymbol{X}_T|\boldsymbol{X}_0, \sigma_{\max}^2\mathbf{I})$. In particular, we need to make sure that $\sigma_{\max}$ is large enough that $\mathcal{N}(\boldsymbol{X}_T|\boldsymbol{X}_0, \sigma_{\max}^2\mathbf{I}) \approx \mathcal{N}(\boldsymbol{X}_T|\mathbf{0}, \sigma_{\max}^2\mathbf{I})$.

However, in practice, lots of diffusion models (Jo et al., 2022; Luo et al., 2024; Wen et al., 2024) adopted a rather conservative strategy when training the network. For VPSDE, results in the maximum forward perturbation distribution being $\mathcal{N}(\boldsymbol{X}_T|u_T\boldsymbol{x}_0, \sigma_T^2\mathbf{I})$, which is far from reaching $\mathcal{N}(\boldsymbol{X}_T|\mathbf{0}, \mathbf{I})$. For VESDE, due to $\sigma_{\max}$ not being large enough, the maximum forward perturbation distribution is $\mathcal{N}(\boldsymbol{X}_T|\boldsymbol{X}_0, \sigma_{\max}^2\mathbf{I})$, which cannot be approximated by $\mathcal{N}(\boldsymbol{X}_T|\mathbf{0}, \sigma_{\max}^2\mathbf{I})$. However, baselines always start reverse sampling from the standard Gaussian distribution, which leads to significant reverse-starting bias. A detailed comparison of parameters is shown in Tables 7, 8, and 9.

## B    FIGURE DETAILS

In this section, we present the detailed procedures to plot Fig. 2. Let $s_{\boldsymbol{\theta},t}(\cdot)$ represent the pretrained GDSS score network. Due to the conservative strategy of GDSS, with $\beta_{\min} = 0.1$ and $\beta_{\max} = 1$,

Table 7: The actual parameters of the forward perturbation of GDSS.

| Model | GDSS | | | | | | | | | |
|---|---|---|---|---|---|---|---|---|---|---|
| Dataset | Community-small | | Enzymes | | Grid | | QM9 | | ZINC250k | |
| Type | Node | Edge | Node | Edge | Node | Edge | Node | Edge | Node | Edge |
| SDE | VPSDE | VPSDE | VPSDE | VESDE | VPSDE | VPSDE | VESDE | VESDE | VPSDE | VESDE |
| $\beta_{\min}|\sigma_{\min}$ | 0.1 | 0.1 | 0.1 | 0.2 | 0.1 | 0.2 | 0.1 | 0.1 | 0.1 | 0.2 |
| $\beta_{\max}|\sigma_{\max}$ | 1.0 | 1.0 | 1.0 | 1.0 | 1.0 | 0.8 | 1.0 | 1.0 | 1.0 | 1.0 |
| $u_T$ | 0.7596 | 0.7596 | 0.7596 | 1.0 | 0.7596 | 0.7788 | 1.0 | 1.0 | 0.7596 | 1.0 |
| $\sigma_T^2$ | 0.4231 | 0.4231 | 0.4231 | 1.0 | 0.4231 | 0.3935 | 1.0 | 1.0 | 0.4231 | 1.0 |

Table 8: The actual parameters of the forward perturbation of HGDM.

| Model | HGDM | | | | | | | | | |
|---|---|---|---|---|---|---|---|---|---|---|
| Dataset | Community-small | | Enzymes | | Grid | | QM9 | | ZINC250k | |
| Type | Node | Edge | Node | Edge | Node | Edge | Node | Edge | Node | Edge |
| SDE | VPSDE | VPSDE | VPSDE | VESDE | VPSDE | VESDE | VPSDE | VESDE | VPSDE | VESDE |
| $\beta_{\min}|\sigma_{\min}$ | 0.1 | 0.1 | 0.1 | 0.2 | 0.1 | 0.2 | 0.1 | 0.1 | 0.1 | 0.2 |
| $\beta_{\max}|\sigma_{\max}$ | 1.0 | 1.0 | 1.0 | 1.0 | 7.0 | 0.8 | 2.0 | 1.0 | 1.0 | 1.0 |
| $u_T$ | 0.7596 | 0.7596 | 0.7596 | 1.0 | 0.1695 | 1.0 | 0.5916 | 1.0 | 0.7596 | 1.0 |
| $\sigma_T^2$ | 0.4231 | 0.4231 | 0.4231 | 1.0 | 0.9713 | 0.64 | 0.6501 | 1.0 | 0.4231 | 1.0 |

Table 9: The actual parameters of the forward perturbation of GSDM and MOOD.

| Model | GSDM | | | | | | | | MOOD | |
|---|---|---|---|---|---|---|---|---|---|---|
| Dataset | Community-small | | Enzymes | | Grid | | QM9 | | ZINC250k | |
| Type | Node | Eigen | Node | Eigen | Node | Eigen | Node | Edge | Node | Edge |
| SDE | VPSDE | VPSDE | VPSDE | VPSDE | VPSDE | VPSDE | VPSDE | VESDE | VPSDE | VESDE |
| $\beta_{\min}|\sigma_{\min}$ | 0.1 | 0.1 | 0.1 | 0.1 | 0.1 | 0.2 | 0.1 | 0.1 | 0.1 | 0.2 |
| $\beta_{\max}|\sigma_{\max}$ | 1.0 | 1.0 | 1.0 | 1.0 | 1.0 | 0.8 | 1.0 | 1.0 | 1.0 | 1.0 |
| $u_T$ | 0.7596 | 0.7596 | 0.7596 | 0.7596 | 0.7596 | 0.7788 | 1.0 | 1.0 | 0.7596 | 1.0 |
| $\sigma_T^2$ | 0.4231 | 0.4231 | 0.4231 | 0.4231 | 0.4231 | 0.3935 | 1.0 | 1.0 | 0.4231 | 1.0 |

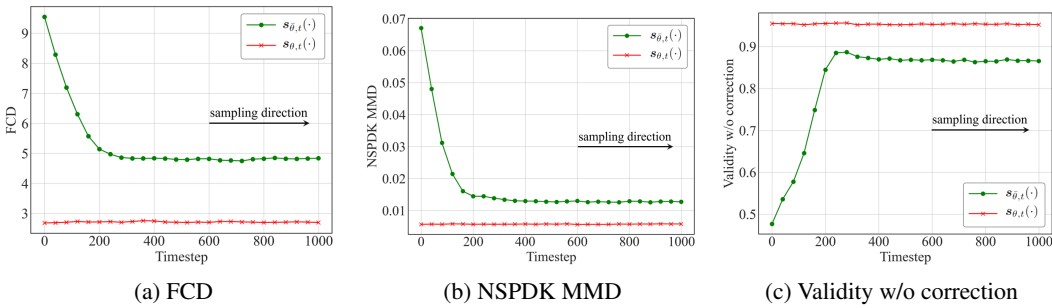

(a) FCD         (b) NSPDK MMD         (c) Validity w/o correction

Figure 4: Generation metric responses to perturbations at different time steps for two score networks.

the maximum perturbation distribution is $\mathcal{N}\left(\boldsymbol{X}_T|0.7596\boldsymbol{X}_0, 0.4231\mathbf{I}\right)$ at $t = T$. On the other hand, $\boldsymbol{s}_{\boldsymbol{\psi},t}(\cdot)$ is defined with the constraints of $\beta_{\min} = 0.1$ and $\beta_{\max} = 20$. At $t = T$, the maximum perturbation distribution is $\mathcal{N}(\boldsymbol{X}_T|\mathbf{0}, \mathbf{I})$. Then, $\boldsymbol{s}_{\boldsymbol{\theta},t}(\cdot)$ and $\boldsymbol{s}_{\boldsymbol{\psi},t}(\cdot)$ not only represent two different networks but also indicate that their maximum perturbation distributions in the forward diffusing are completely different.

To plot Figs. 2(a) and 2(b), we freeze the converged $\boldsymbol{s}_{\boldsymbol{\theta},t}(\cdot)$ and $\boldsymbol{s}_{\boldsymbol{\psi},t}(\cdot)$, then replace $\boldsymbol{X}_t$ in Eq. (3) with $\boldsymbol{A}_t$ to obtain perturbation samples of 1024 edges at different time steps. We then compute $\|\boldsymbol{s}_{\boldsymbol{\theta},t}(\cdot)\|_2$ and $\|\boldsymbol{s}_{\boldsymbol{\psi},t}(\cdot)\|_2$ and plot them in Fig. 2.

To plot Fig. 2(c), we introduce perturbations to $\boldsymbol{s}_{\boldsymbol{\theta},t}(\cdot)$ at different time stepsin sampling. We employ a sampling method without a corrector and perturb the score at the selected time step ($y$-axis) using Gaussian noise:

$$\boldsymbol{s}_{\boldsymbol{\theta},t}(\cdot) = \boldsymbol{s}_{\boldsymbol{\theta},t}(\cdot) + \boldsymbol{z}_t \tag{14}$$

where $\boldsymbol{z}_t \sim \mathcal{N}(\boldsymbol{z}_t|\mathbf{0}, \mathbf{I})$. For the other time steps, we do not introduce any perturbations, allowing the diffusion model to perform sampling and record the generation metrics. We conduct the perturbation experiment on $\boldsymbol{s}_{\boldsymbol{\theta},t}(\cdot)$ using the same method, and ultimately compare the results of the two perturbation experiments based on the time steps to evaluate how different score networks in the diffusion model resist bias at various time steps. We present a detailed comparison of the generation metrics from the perturbation experiments, as shown in Fig. 4.

## C  DERIVATIONS FOR §4.2

For a diffusion model, let the true data be $\boldsymbol{X}_0$, and the pretrained score network be $\boldsymbol{s}_{\boldsymbol{\theta},t}(\cdot)$. Based on the set noise addition method, we have

$$\nabla_{\boldsymbol{X}_t} \log q(\boldsymbol{X}_t|\boldsymbol{X}_0) = -\frac{\boldsymbol{X}_t - \sqrt{\bar{\alpha}_t}\boldsymbol{X}_0}{1 - \bar{\alpha}_t}. \tag{15}$$

However, $\boldsymbol{s}_{\boldsymbol{\theta},t}(\cdot)$ often deviates from the optimal logarithmic gradient of $q(\boldsymbol{X}_t|\boldsymbol{X}_0)$. In the reverse-sampling process, assuming the current time $t$ has a data state $\hat{\boldsymbol{X}}_t$, the predicted score is

$$\boldsymbol{s}_{\boldsymbol{\theta},t}(\hat{\boldsymbol{X}}_t) = -\frac{\hat{\boldsymbol{X}}_t - \sqrt{\bar{\alpha}_t}\hat{\boldsymbol{X}}_0}{1 - \bar{\alpha}_t}. \tag{6}$$

We model $\hat{\boldsymbol{X}}_0$ by

$$\hat{\boldsymbol{X}}_0 = \gamma_t \boldsymbol{X}_0 + \delta_t \boldsymbol{\epsilon}_a. \tag{8}$$

Eq. (6) becomes

$$\boldsymbol{s}_{\boldsymbol{\theta},t}(\hat{\boldsymbol{X}}_t) = -\frac{\hat{\boldsymbol{X}}_t - \sqrt{\bar{\alpha}_t}(\gamma_t \boldsymbol{X}_0 + \delta_t \boldsymbol{\epsilon}_a)}{1 - \bar{\alpha}_t}. \tag{16}$$

Then, we train a new score network $\boldsymbol{s}_{\boldsymbol{\psi},t}(\cdot)$ based on the generated data $X'$ from the score network. Following the above derivation, we can write the predicted score at the current time $t$ as

$$\boldsymbol{s}_{\boldsymbol{\psi},t}(\hat{\boldsymbol{X}}_t) = -\frac{\hat{\boldsymbol{X}}_t - \sqrt{\bar{\alpha}_t}\hat{\boldsymbol{X}}'_0}{1 - \bar{\alpha}_t}. \tag{17}$$

Due to the bias of $s_{\boldsymbol{\theta},t}(\cdot)$, the generated pseudo data $\boldsymbol{X}_0'$ always deviates from $\boldsymbol{X}_0$, and considering the prediction error of the network, we can easily obtain

$$\hat{\boldsymbol{X}}_0' = \gamma_t' \boldsymbol{X}_0 + \delta_t' \boldsymbol{\epsilon}_b \tag{18}$$

where $\gamma_t' < \gamma_t$, $\delta_t' > \delta_t$, meaning that at the same time $t$, $s_{\boldsymbol{\psi},t}(\cdot)$ trained on generated data has a larger bias in predicting the target distribution than $s_{\boldsymbol{\theta},t}(\cdot)$ trained on the true data. By substituting Eq. (18) into Eq. (17), we have

$$s_{\boldsymbol{\psi},t}(\hat{\boldsymbol{X}}_t) = -\frac{\hat{\boldsymbol{X}}_t - \sqrt{\bar{\alpha}_t}(\gamma_t' \boldsymbol{X}_0 + \delta_t' \boldsymbol{\epsilon}_b)}{1 - \bar{\alpha}_t} . \tag{19}$$

The score difference is defined as the difference between Eq. (16) and Eq. (19),

$$
\begin{aligned}
s_{\boldsymbol{\theta},t}(\hat{\boldsymbol{X}}_t) - s_{\boldsymbol{\psi},t}(\hat{\boldsymbol{X}}_t) &= -\frac{\hat{\boldsymbol{X}}_t - \sqrt{\bar{\alpha}_t}(\gamma_t \boldsymbol{X}_0 + \delta_t \boldsymbol{\epsilon}_a)}{1 - \bar{\alpha}_t} - \left( -\frac{\hat{\boldsymbol{X}}_t - \sqrt{\bar{\alpha}_t}(\gamma_t' \boldsymbol{X}_0 + \delta_t' \boldsymbol{\epsilon}_b)}{1 - \bar{\alpha}_t} \right) \\
&= \sqrt{\bar{\alpha}_t} \frac{(\gamma_t - \gamma_t')\boldsymbol{X}_0 + (\delta_t \boldsymbol{\epsilon}_a - \delta_t' \boldsymbol{\epsilon}_b)}{1 - \bar{\alpha}_t} .
\end{aligned}
\tag{9}
$$

We incorporate this score difference as a correction term into the predicted score and introduce a hyperparameter to fine-tune the adjustment, improving the alignment of the predicted score with the true score. It is given by

$$
\begin{aligned}
s_{\boldsymbol{\theta},t}(\hat{\boldsymbol{X}}_t) &= s_{\boldsymbol{\theta},t}(\hat{\boldsymbol{X}}_t) + \lambda\big(s_{\boldsymbol{\theta},t}(\hat{\boldsymbol{X}}_t) - s_{\boldsymbol{\psi},t}(\hat{\boldsymbol{X}}_t)\big) \\
&= -\frac{\hat{\boldsymbol{X}}_t - \sqrt{\bar{\alpha}_t}\big(\gamma_t \boldsymbol{X}_0 + \delta_t \boldsymbol{\epsilon}_a + \lambda(\gamma_t - \gamma_t')\boldsymbol{X}_0 + \lambda(\delta_t \boldsymbol{\epsilon}_a - \delta_t' \boldsymbol{\epsilon}_b)\big)}{1 - \bar{\alpha}_t} \\
&= -\frac{\hat{\boldsymbol{X}}_t - \sqrt{\bar{\alpha}_t}\big(\big(\gamma_t + \lambda(\gamma_t - \gamma_t')\big)\boldsymbol{X}_0 + \delta_t \boldsymbol{\epsilon}_a + \lambda(\delta_t \boldsymbol{\epsilon}_a - \delta_t' \boldsymbol{\epsilon}_b)\big)}{1 - \bar{\alpha}_t} .
\end{aligned}
\tag{10}
$$

Since $\gamma_t' < \gamma_t$, this score difference helps the predicted score incorporate more information from the true data $\boldsymbol{X}_0$. By appropriately setting a hyperparameter $\lambda$, we can consistently use the information from $\boldsymbol{X}_0$ to guide the score correction process. Finally, we introduce an adjustment magnitude coefficient to further refine the score correction based on the score difference, which improves the alignment of the predicted score with the true score.

$$s_{\boldsymbol{\theta},t}(\hat{\boldsymbol{X}}_t) = \frac{s_{\boldsymbol{\theta},t}(\hat{\boldsymbol{X}}_t)}{\omega} . \tag{11}$$

We provide a logical analysis showing that the score difference helps to correct the score.

## D    DETAILS FOR EXPERIMENT

We provide detailed parameters for experiments described in §5, as shown in Tables 10 and 11. Specifically, we distinguish between the relevant parameters for sampling methods with correctors and those without correctors. As shown in Tables 10, both Langevin sampling and the S++ correction are implemented in the reverse-starting alignment stage. After this stage, the subsequent sampling stages require little to no further correction. This suggests that in a truncated diffusion model, once correction is applied at a certain point, the later stages can achieve good sampling performance without the need for additional corrections.

## E    SAMPLING ALGORITHM

In this section, we present the sampling algorithm procedure for S++, as shown in Algorithm 1. $\beta_t = \beta_{\min} + t(\beta_{\max} - \beta_{\min})$ for $t \in [0, 1]$. Additionally, our approach can be naturally integrated into the reverse sampling of various diffusion models without a corrector, greatly improving the generation quality of sampling methods. For methods with a corrector, we can significantly reduce the correction time interval by introducing a truncation time. Specifically, we apply the corrector when the time exceeds $t_c$ further reducing the computational cost.

---

**Algorithm 1** The S++ sampling algorithm.

---

**Input:** An inference model $s_{\boldsymbol{\theta},t}(\cdot)$, a pretrained pseudo model $s_{\boldsymbol{\psi},t}(\cdot)$, and the cut-off time $t_c$.
**Initialize:** $\boldsymbol{X}_T \sim \mathcal{N}(\mathbf{0}, \mathbf{I})$ where $T = \frac{N-1}{N}$ ;
**for** $j = 1$ **to** $M$ **do**

$\quad s_{\boldsymbol{\theta},T}(\boldsymbol{X}_T) \leftarrow \frac{s_{\boldsymbol{\theta},T}(\boldsymbol{X}_T) + \lambda_1\left(s_{\boldsymbol{\theta},T}(\boldsymbol{X}_T) - s_{\boldsymbol{\psi},T}(\boldsymbol{X}_T)\right)}{\omega_1}$ ;
$\quad \boldsymbol{\epsilon} \sim \mathcal{N}(\mathbf{0}, \mathbf{I})$ ;
$\quad \boldsymbol{X}_T \leftarrow \boldsymbol{X}_T + \frac{\eta_T}{2} s_{\boldsymbol{\theta},T}(\boldsymbol{X}_T) + \sqrt{\eta_T}\boldsymbol{\epsilon}$ ;
**end for**
**for** $i = N - 1$ **to** $0$ **do**
$\quad t = \frac{i}{N}$ ;
$\quad s_{\boldsymbol{\theta},t}(\boldsymbol{X}_t) \leftarrow \frac{s_{\boldsymbol{\theta},t}(\boldsymbol{X}_t) + \lambda_2\left(s_{\boldsymbol{\theta},t}(\boldsymbol{X}_t) - s_{\boldsymbol{\psi},t}(\boldsymbol{X}_t)\right)}{\omega_2}$ ;
$\quad \boldsymbol{\epsilon} \sim \mathcal{N}(\mathbf{0}, \mathbf{I})$ ;
$\quad \boldsymbol{X}_t \leftarrow (2 - \sqrt{1 - \beta_t})\boldsymbol{X}_t + \beta_t s_{\boldsymbol{\theta},t}(\boldsymbol{X}_t) + \sqrt{\beta_t}\boldsymbol{\epsilon}$ ;
$\quad$**if** $t \le t_c$ **then**
$\quad\quad$**if** corrector **then**
$\quad\quad\quad \boldsymbol{\epsilon} \sim \mathcal{N}(\mathbf{0}, \mathbf{I})$ ;
$\quad\quad\quad \boldsymbol{X}_t \leftarrow \boldsymbol{X}_t + \frac{\eta_t}{2} s_{\boldsymbol{\theta},t}(\boldsymbol{X}_t) + \sqrt{\eta_t}\boldsymbol{\epsilon}$ ;
$\quad\quad$**end if**
$\quad$**end if**
**end for**
**return** $\boldsymbol{X}_0$ .

---

Table 10: Experimental parameters for sampling methods without a corrector (OC).

| Model | Hyper. | Comm. | Enzymes | Grid | QM9 | ZINC250k |
|---|---|---|---|---|---|---|
| | $M$ | 400 | 420 | 350 | 400 | 400 |
| | $\lambda_1$ | 0.2 | 0.0008 | 0.06 | 1.19 | 2.5 |
| GDSS-OC-S++ | $\omega_1$ | 0.998 | 1.0 | 1.0 | 1.09 | 1.0 |
| | $\lambda_2$ | 0 | 0 | 0 | 0 | 0 |
| | $\omega_2$ | 1.0 | 1.0 | 1.0 | 1.0 | 1.0 |
| | $M$ | 280 | 310 | 280 | 240 | 220 |
| | $\lambda_1$ | 0.02 | 0.0 | 0.02 | 0.1 | 0.025 |
| HGDM-OC-S++ | $\omega_1$ | 1.0 | 1.0 | 1.0 | 1.0 | 1.07 |
| | $\lambda_2$ | 0.36 | 0.0 | 0.0 | 0.36 | 0.0 |
| | $\omega_2$ | 1.0 | 1.0 | 1.0 | 0.78 | 1.0 |
| | $M$ | 200 | 400 | 400 | - | - |
| | $\lambda_1$ | 0.0 | 0.0 | 0.0 | - | - |
| GSDM-OC-S++ | $\omega_1$ | 1.0 | 1.0 | 1.0 | - | - |
| | $\lambda_2$ | 0.0 | 0.0 | 0.0 | - | - |
| | $\omega_2$ | 1.0 | 1.0 | 1.0 | - | - |

## F  ADDITIONAL EXPERIMENTS

To demonstrate the superiority of S++, we select generative models other than diffusion models as baseline models for comparison. GraphVAE (Simonovsky & Komodakis, 2018) is a graph generative model based on variational autoencoders; DeepGMG (Li et al., 2018) is a deep generative model that generates graphs in a sequential, node-by-node manner; GraphAF (Shi et al., 2020) is an autoregressive flow-based model. GraphRNN (You et al., 2018) is an autoregressive model using recurrent neural networks to generate graphs; EDP-GNN (Niu et al., 2020) is a score-based generative model using energy-based dynamics. GraphEBM (Liu et al., 2021) is an energy-based generative model that generates molecules by minimizing energy through Langevin dynamics, which is categorized as a one-shot generative method. We provide detailed comparative experiments in Tables 12 and 13, and the results show that our approach significantly outperforms the baseline models and other generative models.

Table 11: Experimental parameters for sampling methods with a corrector (WC).

| Model | Hyper. | Comm. | Enzymes | Grid | QM9 | ZINC250k |
|---|---|---|---|---|---|---|
| GDSS-WC-S++ | $M$ | 400 | 420 | 350 | 400 | 400 |
| | $\lambda_1$ | 0.2 | 0.0008 | 0.06 | 1.19 | 2.5 |
| | $\omega_1$ | 0.998 | 1.0 | 1.0 | 1.09 | 1.0 |
| | $t_c$ | 0.2 | 0.45 | 0.055 | 0.1 | 0.4 |
| | $\lambda_2$ | 0 | 0 | 0 | 0 | 0 |
| | $\omega_2$ | 1.0 | 1.0 | 1.0 | 1.0 | 1.0 |
| HGDM-WC-S++ | $M$ | 280 | 200 | 360 | 240 | 220 |
| | $\lambda_1$ | 0.02 | 0.0 | 0.18 | 0.1 | 0.25 |
| | $\omega_1$ | 1.0 | 1.0 | 1.0 | 1.0 | 1.07 |
| | $t_c$ | 0.2 | 0.5 | 0.1 | 0.65 | 0.6 |
| | $\lambda_2$ | 0.36 | 0.0 | 0.0 | 0.0 | 0.0 |
| | $\omega_2$ | 1.0 | 1.0 | 1.0 | 1.44 | 0.87 |
| GSDM-WC-S++ | $M$ | 200 | 400 | 400 | - | - |
| | $\lambda_1$ | 0.0 | 0.0 | 0.0 | - | - |
| | $\omega_1$ | 1.0 | 1.0 | 1.0 | - | - |
| | $t_c$ | 0.05 | 0.70 | 0.45 | - | - |
| | $\lambda_2$ | 0.0 | 0.0 | 0.0 | - | - |
| | $\omega_2$ | 1.0 | 1.0 | 1.0 | - | - |

Table 12: Additional experiments on generic graph datasets.

| Dataset Info. | Community-small Synthetic, $12 \leq |V| \leq 20$ | | | | Enzymes Real, $10 \leq |V| \leq 125$ | | | | Grid Synthetic, $100 \leq |V| \leq 400$ | | | |
|---|---|---|---|---|---|---|---|---|---|---|---|---|
| Method | Deg. | Clus. | Orbit | Avg. | Deg. | Clus. | Orbit | Avg. | Deg. | Clus. | Orbit | Avg. |
| DeepGMG | 0.220 | 0.950 | 0.400 | 0.523 | - | - | - | - | - | - | - | - |
| GraphRNN | 0.080 | 0.120 | 0.040 | 0.080 | 0.017 | 0.062 | 0.046 | 0.042 | 0.064 | 0.043 | 0.021 | 0.043 |
| GraphAF | 0.18 | 0.20 | 0.02 | 0.133 | 1.669 | 1.283 | 0.266 | 1.073 | - | - | - | - |
| GraphDF | 0.06 | 0.12 | 0.03 | 0.070 | 1.503 | 1.061 | 0.202 | 0.922 | - | - | - | - |
| GraphVAE | 0.350 | 0.980 | 0.540 | 0.623 | 1.369 | 0.629 | 0.191 | 0.730 | 1.619 | 0.0 | 0.919 | 0.846 |
| EDP-GNN | 0.053 | 0.144 | 0.026 | 0.074 | 0.023 | 0.268 | 0.082 | 0.124 | 0.455 | 0.238 | 0.328 | 0.340 |
| GDSS-OC | 0.050 | 0.132 | 0.011 | 0.064 | 0.052 | 0.627 | 0.249 | 0.309 | 0.270 | 0.009 | 0.034 | 0.070 |
| GDSS-OC-S++ | **0.021** | **0.061** | **0.005** | **0.029** | **0.067** | **0.099** | **0.007** | **0.058** | **0.105** | **0.004** | **0.061** | **0.066** |
| GDSS-WC | 0.045 | 0.088 | 0.007 | 0.045 | 0.044 | 0.069 | 0.002 | 0.038 | 0.111 | 0.005 | 0.070 | 0.070 |
| GDSS-WC-S++ | **0.019** | **0.062** | **0.004** | **0.028** | **0.031** | **0.050** | **0.003** | **0.028** | **0.105** | **0.004** | **0.061** | **0.057** |
| HGDM-OC | 0.065 | 0.119 | 0.024 | 0.069 | 0.125 | 0.625 | 0.371 | 0.374 | 0.181 | 0.019 | 0.112 | 0.104 |
| HGDM-OC-S++ | **0.021** | **0.034** | **0.005** | **0.020** | **0.080** | **0.500** | **0.225** | **0.268** | **0.023** | **0.034** | **0.004** | **0.020** |
| HGDM-WC | 0.017 | 0.050 | 0.005 | 0.024 | 0.045 | 0.049 | 0.003 | 0.035 | 0.137 | 0.004 | 0.048 | 0.069 |
| HGDM-WC-S++ | **0.021** | **0.024** | **0.004** | **0.016** | **0.040** | **0.041** | **0.005** | **0.029** | **0.123** | **0.003** | **0.047** | **0.058** |
| GSDM-OC | 0.142 | 0.230 | 0.043 | 0.138 | 0.930 | 0.867 | 0.168 | 0.655 | 1.996 | 0.0 | 1.013 | 1.003 |
| GSDM-OC-S++ | **0.011** | **0.016** | **0.001** | **0.009** | **0.012** | **0.087** | **0.011** | **0.037** | **1.2e-4** | **0.0** | **1.2e-4** | **0.066** |
| GSDM-WC | 0.011 | 0.016 | 0.001 | 0.009 | 0.013 | 0.088 | 0.013 | 0.038 | 0.002 | 0.0 | 0 | 7.2e-5 |
| GSDM-WC-S++ | **0.011** | **0.016** | **0.001** | **0.009** | **0.011** | **0.086** | **0.010** | **0.036** | **5.0e-5** | **0.0** | **1.1e-5** | **0.066** |

Table 13: Additional experiments on QM9 and ZINC250k datasets.

| Method | QM9 | | | ZINC250k | | |
|---|---|---|---|---|---|---|
| | Val. w/o corr. (%)↑ | NSPDK MMD ↓ | FCD ↓ | Val. w/o corr. (%)↑ | NSPDK MMD ↓ | FCD ↓ |
| GraphAF | 67.00 | 0.020 | 5.268 | 68.00 | 0.044 | 16.289 |
| GraphDF | 82.67 | 0.063 | 10.816 | 89.03 | 0.176 | 34.202 |
| MoFlow | 91.36 | 0.017 | 4.467 | 63.11 | 0.046 | 20.931 |
| EDP-GNN | 47.52 | 0.005 | 2.680 | 82.97 | 0.049 | 16.737 |
| GraphEBM | 8.22 | 0.030 | 6.143 | 5.29 | 0.212 | 35.471 |
| GDSS-OC | 73.49 | 0.015 | 4.584 | 41.84 | 0.047 | 20.53 |
| GDSS-OC-S++ | **93.74** | **0.001** | **1.661** | **59.50** | **0.050** | **16.79** |
| GDSS-WC | 94.91 | 0.004 | 2.550 | 95.83 | 0.019 | 14.66 |
| GDSS-WC-S++ | **93.79** | **0.001** | **1.661** | **93.15** | **0.012** | **12.70** |
| HGDM-OC | 92.22 | 0.005 | 3.164 | 66.47 | 0.033 | 21.38 |
| HGDM-OC-S++ | **94.95** | **0.003** | **2.512** | **67.12** | **0.034** | **20.79** |
| HGDM-WC | 98.02 | 0.002 | 2.147 | 93.26 | 0.016 | 17.69 |
| HGDM-WC-S++ | **97.03** | **0.001** | **2.001** | **91.03** | **0.016** | **16.24** |

Table 14: FCD($\downarrow$) on GDSS-OC baseline and QM9 under different parameters $\lambda$ ($\lambda = 0$ represents the baseline).

| $\lambda$ | 0 | 1.17 | 1.18 | 1.19 | 1.20 | 1.21 |
|---|---|---|---|---|---|---|
| FCD | 4.583 | 1.768 | 1.756 | 1.754 | 1.761 | 1.762 |

Table 15: Degree($\downarrow$) on GDSS-OC baseline and Community-small under different parameters $\lambda$ ($\lambda = 0$ represents the baseline).

| $\lambda$ | 0 | 0.18 | 0.19 | 0.20 | 0.21 | 0.22 |
|---|---|---|---|---|---|---|
| Degree | 0.05 | 0.023 | 0.024 | 0.022 | 0.024 | 0.026 |

## G   INSENSITIVITY OF $\lambda$

We emphasize that S++ is largely insensitive to the choice of $\lambda$, as performance improvements are consistently observed across a wide range of $\lambda$ values, as shown in Tables 14 and 15.

## H   TWO BIASES IN IMAGE AND GRAPH

Given the existing research on reverse-starting bias and exposure bias in the image domain, this section highlights the key differences between these biases in image and graph data, focusing on three aspects: data scale, data structure, and network performance.

**1-a Reverse-starting bias in images.** DPM-Fixes (Lin et al., 2024) was the first to identify that traditional noise scheduling strategies fail to ensure the maximum forward perturbation distribution aligns with a standard Gaussian distribution:

$$\boldsymbol{x}_T = 0.068265\boldsymbol{x}_0 + 0.997667\boldsymbol{\epsilon}_T \,. \tag{20}$$

This shows a slight deviation from the standard Gaussian starting point used in reverse sampling. To address this, DPM-Fixes constrains the forward $\boldsymbol{x}_T$ to follow the standard Gaussian by adjusting the noise scheduling scale. This approach benefits from large image datasets and robust network performance, which enable accurate noise (or score) predictions even in high-noise states. On the other hand, DPM-Leak (Everaert et al., 2024) estimates the maximum forward perturbation distribution during diffusion based on pixel modeling, using it as the new reverse-starting point for sampling. This strategy capitalizes on the image data structure and scale, which allows pixels to be modeled independently and follow a Gaussian distribution.

**1-b Reverse-starting bias in graphs.** Unlike image data, where the models can rely on more manageable data scales and network capacities, graph models are hindered by the challenges of accurately predicting noise (or score) from high-noise states. As a result, baseline models tend to adopt a conservative strategy during training, leading to the maximum forward perturbation distribution significantly deviating from a standard Gaussian. While this approach avoids the instability of high-noise states, it introduces considerable reverse-starting bias. Consequently, strategies like DPM-Fixes, which enforce $\boldsymbol{x}_T$ to follow a standard Gaussian distribution, are not suitable. Given the interdependencies of nodes and edges in graph data and its inherent sparsity, we cannot assume that nodes or edges follow a Gaussian distribution to estimate the maximum forward perturbation. This makes approaches like DPM-Leak equally inappropriate.

**2-a Exposure bias in images.** In images, exposure bias refers to the mismatch between forward process $\boldsymbol{x}_t$ and reverse process $\hat{\boldsymbol{x}}_t$, with differences accumulating throughout sampling, ultimately affecting generation quality. Many current image exposure bias works assume no reverse-starting bias exists, focusing on sampling process bias, as reverse-starting bias in images is indeed quite minimal.

**2-b Exposure bias in graphs.** Unlike the minor signal leakage seen in image diffusion models, graph diffusion models suffer from significant reverse-starting bias, leading to substantial exposure bias after the initial sampling step. In other words, the exposure bias in graphs is not solely a result of network prediction errors or the accumulation of sampling iterations, but is heavily influenced by the reverse-starting bias. Therefore, effectively addressing graph exposure bias requires first mitigating the reverse-starting bias.

In conclusion, we emphasize that reverse-starting bias is a particularly acute and unique issue in graph diffusion models. While exposure bias also affects graphs, solutions developed for image-based models cannot be directly applied. This work is the first to focus specifically on reverse-starting bias in graph diffusion models, proposing a simple yet effective solution.

Table 16: FCD($\downarrow$) on QM9 without a corrector.

| $\omega$ | 0.7 | 0.8 | 0.9 | 1.0 | 1.1 |
|---|---|---|---|---|---|
| GDSS-OC-ES | 3.57 | 3.417 | 3.768 | 4.584 | 5.517 |
| GDSS-WC-S++ | **2.94** | **2.814** | **3.198** | **4.187** | **5.201** |

Table 17: FCD($\downarrow$) on QM9 with a corrector.

| $\omega$ | 0.9 | 1.0 | 1.1 | 1.2 | 1.3 |
|---|---|---|---|---|---|
| GDSS-WC-ES | 2.809 | 2.552 | 2.319 | 2.301 | 2.542 |
| GDSS-WC-S++ | **2.321** | **2.034** | **1.858** | **1.852** | **2.152** |

## I S++ AND EXISTING SOLUTIONS IN IMAGES

Although there are many solutions to mitigate the exposure bias in current images, these solutions cannot replace S++. In this section, we choose to compare S++ in detail with similar works (Li et al., 2024; Ning et al., 2024), as they are plug-and-play solutions that do not introduce new components.

TS-DPM (Li et al., 2024) proposes searching an optimal time $s$ during sampling. TS-DPM relies on two fundamental assumptions: (a) image pixels are independent and follow Gaussian distribution, Eq. (13) in Appendix J of Li et al. (2024); (b) sample pixel variance approximates population variance, Eq. (20) in Appendix J of Li et al. (2024). These assumptions are based on large image datasets and a large number of pixels. However, nodes and edges in graphs are highly sparse. Specifically, many graph datasets are small (Community-small, Enzymes, and Grid have fewer than 1000 samples). Both assumptions do not hold for graph data, making TS-DPM inapplicable to graph diffusion models.

ADM-ES(Ning et al., 2024) proposes reducing $s_{\boldsymbol{\theta},t}(\cdot)$, originally noise $\epsilon_{\boldsymbol{\theta},t}(\cdot)$, during sampling to mitigate exposure bias. However, this approach does not involve the angle of $s_{\boldsymbol{\theta},t}(\cdot)$. Our approach addresses this limitation:

$$s_{\boldsymbol{\theta},t}(\boldsymbol{X}_t) = \frac{s_{\boldsymbol{\theta},t}(\boldsymbol{X}_t) + \lambda\big(s_{\boldsymbol{\theta},t}(\boldsymbol{X}_t) - s_{\boldsymbol{\psi},t}(\boldsymbol{X}_t)\big)}{\omega} \ . \tag{21}$$

If $\lambda = 0$, it is equivalent to ADM-ES. In other words, ADM-ES is a special case of S++, while the score difference $s_{\boldsymbol{\theta},t}(\boldsymbol{X}_t) - s_{\boldsymbol{\psi},t}(\boldsymbol{X}_t)$ provides the direction information.

For fair comparison with ADM-ES, we introduce $\lambda\big(s_{\boldsymbol{\theta},t}(\boldsymbol{X}_t) - s_{\boldsymbol{\psi},t}(\boldsymbol{X}_t)\big)$ for each magnitude factor $\omega$, with $\lambda$ uniformly set to 0.5, to examine whether $s_{\boldsymbol{\theta},t}(\boldsymbol{X}_t) - s_{\boldsymbol{\psi},t}(\boldsymbol{X}_t)$ brings improvements over ADM-ES. Tables 16 and 17 demonstrate that the angle information in S++ leads to significant gains.

