# OpenReview forum: "Bias Mitigation in Graph Diffusion Models"
_ICLR.cc/2025/Conference — ICLR 2025 Poster_

### Official Review · Reviewer_S73B · 2024-10-28

**Soundness:** 3
**Presentation:** 3
**Contribution:** 2
**Rating:** 6
**Confidence:** 3

**Summary:**

To solve the exposure bias and the reverse starting bias at the same time, the authors propose to adopt Langevin sampling to obtain samples aligned with the forward maximum perturbation distribution. A fraction correction mechanism is presented. It is based on score difference to solve the exposure bias of the fraction network. The approach is free of network modification and introduction of new components. Empirical experiments demonstrate that the proposed method can achieve state-of-the-art performance on multiple datasets and multiple tasks.

**Strengths:**

Contribution: This paper aims at mitigating the reverse starting bias and the exposure bias at the same time, which is obviously an important topic that yet to be solved.

Solid and sound method: The proposed fraction correction mechanism is elaborated clearly. Multiple medias are employed to help with understanding the methods.

Ample empirical analysis: The proposed method is compared with multiple modern baselines on several datasets. The experiment results are presented clearly. Ablation tests are provided to demonstrate the importance of S++.

**Weaknesses:**

Novelty: Langevin sampling has long been recognized as one of the best approach to learn the distribution in diffusion models. This paper adapts it to the graph diffusion modeling task.

Experiment: The environment of the experiment is not specified.

Acceleration measurement: S++ is claimed to be able to generate good samples faster by using fewer steps of reverse diffusion. Experiment results showed that it could achieve satisfying samples in fewer steps. However, whether it costs more time in each step due to the extra computational burden is not discussed.

**Questions:**

Can you elaborate more about how graph diffusion modeling is different from the general diffusion modeling regarding the two bias?

---

### Official Review · Reviewer_bDyB · 2024-11-01

**Soundness:** 2
**Presentation:** 2
**Contribution:** 3
**Rating:** 6
**Confidence:** 3

**Summary:**

The authors focused on two problems in the Diffusion models for graph data. The first problem is the reverse starting point bias, in which the maximally perturbed data distribution is far from the standard normal distribution from which the reverse process starts during the inference. The second problem is the exposure bias, in which  diffusion model generates data with errors that accumulate across every step of the reverse process, leading to a large gap between the generated data and the true data distribution. The paper proposed solutions to the two problems respectively, and the empirical results are provided to verify the effectiveness of the methods.

**Strengths:**

The paper is well motivated and explained the intuitive and mathematical explanations of the proposed solutions to the problems. In particular, the solution based on the Langevin sampling to the first problem sounds well and is novel.

**Weaknesses:**

**Main arguments**:

- *A1*: Two solutions are proposed in the paper, which are not synergetic and not specific to graph diffusion models (e.g., they can be applied to other types of data). While this is a another strength, given its generic nature, the paper needs to position itself in the existing literature, in particular in context of the second problem (exposure bias). For example, some solutions to the exposure bias exists [1,2] for image generation, and they can be replaced with the proposed solution to the exposure bias. How effective is the proposed solution compared to the existing ones is not clear in the paper, questioning the significance of the solution.
  1. Li, Mingxiao, et al. "Alleviating exposure bias in diffusion models through sampling with shifted time steps." arXiv preprint arXiv:2305.15583 (2023).
  2. Ning, Mang, et al. "Elucidating the exposure bias in diffusion models." arXiv preprint arXiv:2308.15321 (2023).

- *A2*: The solution to the exposure bias has a key parameter, $\lambda$, that controls the degree of correction during inference. In the experiment, the authors used specific value of $\lambda$. However, it is not clear how did they choose and whether the method is sensitive to the choice of $\lambda$. If  $\lambda$ is a highly sensitive parameter, the practical value of the method is limited.

Given the above arguments, I believe that the paper is not strong enough to be accepted at ICLR 2025 at its current form.

**Questions:**

1. What is the key features and novelty of the proposed exposure bias correction in light of the existing methods?
2. How was the $\lambda$ selected in the experiment? Why do they differ across different experiments? How sensitive the results are to the choice of $\lambda$ values?

---

### Official Review · Reviewer_fe9F · 2024-11-02

**Soundness:** 3
**Presentation:** 3
**Contribution:** 3
**Rating:** 8
**Confidence:** 3

**Summary:**

This paper improves GDSS  from a unified perspective and solves a significant problem of exposure bias during graph sampling.
Their approach requires no network modifications, which is validated across multiple models, datasets, and tasks compared with SOTA methods.

**Strengths:**

1. The proposed method effectively mitigates reverse starting bias by employing a newly designed Langevin sampling algorithm.

2. It introduces a fraction correction mechanism based on a newly defined score difference to address exposure bias.

3. The approach requires no network modifications and demonstrates state-of-the-art performance across multiple models, datasets, and tasks.

**Weaknesses:**

For me, I understand the problems the author wants to solve, and they are indeed meaningful (if solved). But I think the biggest problem is that the paper's presentation is not satisfactory, so I cannot give it a higher score. Perhaps, in the rebuttal process, reasonable explanations can alleviate my bias.

**Questions:**

1. Can the proposed model generate attributes?

2. The authors repeatedly emphasize that "Their approach requires no network modification." My confusion is that the noise or bias will not be the same for different networks. Suppose the proposed approach is robust to the network and does not require modifications in different scenarios. How can it ensure that the new reverse starting point is robust and correct across different noises in different networks?

3. The authors claim that they can achieve good performance across multiple tasks and have strong reusability. However, since the code is not open-sourced, I am unable to evaluate it.

4. The abstract is very intriguing, but the introduction is somewhat difficult to read. For example, some symbol definitions and descriptions are not explained, and the authors assume that this knowledge is already known to readers. Additionally, I did not fully understand how the authors addressed the existing problems in Q1 and Q2, as this was not clearly explained.

---

### Official Review · Reviewer_wgRC · 2024-11-04

**Soundness:** 3
**Presentation:** 3
**Contribution:** 3
**Rating:** 6
**Confidence:** 3

**Summary:**

In view that the existing graph diffusion models can not reach the standard Gaussian distribution if following their defined transition distribution, this paper proposes a Langevin sampling algorithm to align with the forward maximum perturbation distribution. Extensive experiments have verified the effectiveness of the proposed method in generating better graphs.

**Strengths:**

(1) The observed misalignment between the maximum perturbation distribution and the actual starting distribution in the generation phase is very important and novel.

(2) The proposed technique for correcting the misalignment bias is technically sound. Furthermore, Figure 1 has demonstrated its effectiveness.

(3) Extensive experiments have been conducted to verify the effectiveness of the proposed method.

**Weaknesses:**

(1) Some of the motivations are not so clear based on the experiments. See question in (3).

**Questions:**

(1) In line 87, the author mentions that the proposed bias correction method can be integrated into existing methods (e.g., spatial, spectral, and hyperbolic domains). I wonder if it also includes discrete graph diffusion, such as DiGress, since that one proves to be more effective in generating discrete graph structure.

(2) In line 205, how do we get the score function $s_{\bar{\theta}, t}(\cdot)$? Is it $s_{\phi, t}(\cdot)$ based on the Appendix B?

(3) I do not fully understand the insights drawn between line 232 and line 236. Why do these experiments provide these two directions for addressing the reverse sampling bias?

(4) How is the continuous diffusion model equipped with the proposed strategy here compared with the discrete diffusion model such as GDSS?

(5) In Figure 2(c), why the perturbation at the very early stage does not lead to any tweaking impact in $s_{\theta, t}(\cdot)$?

---

### Meta-Review · Area_Chair_jSCZ · 2024-12-23

**Metareview:**

The paper proposes a solution to exposure bias and the reverse starting bias in diffusion models using Langevin sampling to obtain samples aligned with the forward maximum perturbation distribution. All the reviewers appreciated the contribution and, during the rebuttal period, they became persuaded by the arguments made by the authors regarding their original concerns/questions. As a consequence, I recommend acceptance.

**Additional Comments On Reviewer Discussion:**

The authors were able to persuade several of the reviewers to increase their original score during the rebuttal period.

---

### Decision · Program_Chairs · 2025-01-22

Accept (Poster)